# HPF1 remodels the active site of PARP1 to enable the serine ADP-ribosylation of histones

Fa-Hui Sun[1,2,3,6], Peng Zhao[1,2,3,6], Nan Zhang[4,5], Lu-Lu Kong[1,2,3], Catherine C. L. Wong [4] &
Cai-Hong Yun [1,2,3✉]

Upon binding to DNA breaks, poly(ADP-ribose) polymerase 1 (PARP1) ADP-ribosylates itself and other factors to initiate DNA repair. Serine is the major residue for ADP-ribosylation upon DNA damage, which strictly depends on HPF1. Here, we report the crystal structures of human HPF1/PARP1-CAT ΔHD complex at 1.98 Å resolution, and mouse and human HPF1 at 1.71 Å and 1.57 Å resolution, respectively. Our structures and mutagenesis data confirm that the structural insights obtained in a recent HPF1/PARP2 study by Suskiewicz et al. apply to PARP1. Moreover, we quantitatively characterize the key residues necessary for HPF1/PARP1 binding. Our data show that through salt-bridging to Glu284/Asp286, Arg239 positions Glu284 to catalyze serine ADP-ribosylation, maintains the local conformation of HPF1 to limit PARP1 automodification, and facilitates HPF1/PARP1 binding by neutralizing the negative charge of Glu284. These findings, along with the high-resolution structural data, may facilitate drug discovery targeting PARP1.

---

[1] Department of Biochemistry and Biophysics, Peking University Health Science Center, Beijing, China. [2] Department of Integration of Chinese and Western Medicine, Peking University Health Science Center, Beijing, China. [3] Beijing Key Laboratory of Tumor Systems Biology, School of Basic Medical Sciences, Peking University Health Science Center, Beijing, China. [4] School of Basic Medical Sciences, Peking University Health Science Center, Beijing, China. [5] Center for Precision Medicine Multi-omics Research, State Key Laboratory of Natural and Biomimetic Drugs, School of Pharmaceutical Sciences, Peking-Tsinghua Center for Life Sciences, Peking University, Beijing, China. [6] These authors contributed equally: Fa-Hui Sun, Peng Zhao. ✉email: yunch@pku.edu.cn

C ovalently attaching the ADP-ribose moiety from NAD$^+$ to substrate molecules (ADP-ribosylation, ADPr) is a reversible post-translational modification involved in various biological processes, including DNA damage response (DDR), DNA replication, transcription, chromatin modulation, host-pathogen interactions, RNA metabolism, unfolded protein response, and mitosis[1–4]. The poly(ADP-ribose) polymerases (PARPs) family, also known as ADP-ribosyltransferase diphtheria toxin-like proteins (ARTDs), are the major enzymes catalyzing ADPr in eukaryotes[5]. Although 17 PARP family members have been discovered, 80–90% of cellular NAD$^+$ consumption during DDR is related to the founding member PARP1, which plays an essential role in DNA damage repair[6,7]. When DNA damage occurs, PARP1 can sense and attach to DNA chain breaks in milliseconds, which in turn activates its ADP-ribosyl transferase activity for modification of itself or other factors, including histones. The electronegative modifications further recruit other DDR factors to launch DNA repair. Due to its extensive and pivotal role in DDR, PARP1 has emerged as a promising target for cancer treatment[8–12]. Because of PARP1's relevance in many tumor types, this is a highly active field of research. Since 2014, several PARP1 inhibitors based on the synthetic lethality strategy (such as olaparib and rucaparib) have been approved for the treatment of ovarian cancer, prostate cancer, and breast cancer, and more inhibitors are being developed[13–15].

PARP1 consists of six domains[16,17] (Supplementary Fig. 1). Three zinc finger domains (Zn1, Zn2 and Zn3) are responsible for recognizing and binding to DNA breaks. The automodification domain (AD) contains a BRCA-C-terminus (BRCT) fold and bears multiple poly-ADP-ribosylation sites. The Trp-Gly-Arg (WGR) domain interacts with Zn1, Zn3, the catalytic domain (CAT), and DNA, and is essential for coupling DNA binding to ADP-ribosyl transferase activation. CAT contains two subdomains: the helical domain (HD) and the ADP-ribosyltransferase (ART) domain. HD acts as an autoinhibitory domain in the folded state, and undergoes local unfolding upon PARP1 activation to enable NAD$^+$ binding and enzyme activation, while the ART domain catalyzes the transfer of ADP-ribose and is highly conserved in other ADP-ribosyl transferases (ARTs). In the last decade, several studies elegantly demonstrated that an inter-domain interaction network is induced by the binding of DNA breaks to the zinc finger domains, which triggers conformational changes of the catalytic domain, especially the local unfolding of HD followed by complete exposure of the NAD$^+$-binding pocket in the catalytic core, leading to PARP1 activation[18–22].

In addition to understanding the intramolecular activation mechanism of PARP1, it is also necessary to clarify how other factors modulate the enzymatic function of PARP1. A previously uncharacterized protein, C4orf27, was demonstrated to be a cofactor of PARP1. C4orf27 physically interacts with PARP1, restricts its hyper-automodification, promotes histone modification, and changes the amino acid specificity of ADP-ribosylation from aspartate/glutamate to serine[23–25]. This protein was therefore named histone PARylation factor 1 (HPF1). Soon afterwards, the serine residue was proved to be the major target of PARP1 upon DDR[26]. These studies implied an important role of HPF1 in regulating PARP1 function. Recently, Suskiewicz et al. determined the crystal structures of *Nematostella vectensis* and *Homo sapiens* HPF1 (both at 2.09 Å resolution), and *Homo sapiens* HPF1/ PARP2-CAT ΔHD complex (at 2.96 Å resolution), which provided important insights into how the binding of HPF1 to PARP2 promotes serine ADPr[27]. However, a high-resolution structure of HPF1 in complex with PARP1, the most important member of the PARPs family, was unavailable. Some unresolved questions related to the assembly and function of the complex also remain. For example, why is Arg239 indispensable for the interaction between

HPF1 and PARP1/2 when this residue does not interact with any residue of PARP1/2? What functional roles does the conserved HPF1 Arg239 residue play? To answer these questions, we determined the crystal structures of mouse and human HPF1, and human HPF1/PARP1 complex at 1.71 Å, 1.57 Å, and 1.98 Å resolution, respectively. In addition, we studied the function of the key residues participating in the HPF1/PARP1 interaction through extensive site-directed mutagenesis, ADPr activity assays, isothermal titration calorimetry (ITC), and mass spectrometry. Our work sheds light on the hitherto obscure role of Arg239 in regulating complex assembly and function, while the high resolution HPF1/PARP1 complex structure may facilitate the design of drugs for this important target in the future.

## Results

**HPF1 binds to the activated ART domain of PARP1**. HPF1 has been shown to bind to the CAT domain (HD-ART) of PARP1 in response to DNA damage[23]. Since HD is an autoinhibitory domain that blocks productive binding to NAD$^+$ in the resting state and undergoes local unfolding to enable NAD$^+$ binding and ADP-ribose transferase activity when PARP1 binds to DNA breaks[19], we asked if unfolding or removing the HD subdomain is the prerequisite for HPF1 binding to PARP1. To answer this question, we tested HPF1 binding to the PARP1 CAT domain in the autoinhibited form (residues 661–1014, full-length CAT), as well as in the constitutively activated form by removing the majority of the autoinhibitory domain HD (CAT ΔHD, missing residues 679–786) using size-exclusion chromatography (SEC). Interestingly, it was observed that CAT ΔHD, but not full-length CAT, forms a complex with HPF1 (Fig. 1a). The interaction was further quantitatively characterized by ITC, which showed that CAT ΔHD binds to HPF1 with a dissociation constant ($K_d$) of ~1.5 $\mu$M and 1:1 stoichiometry (Fig. 1b, Table 1, Supplementary Figs. 2 and 3), while full-length CAT showed no binding to HPF1 (Fig. 1c). Since these observations confirmed that HPF1 binds to activated CAT, we also tested the binding of HPF1 to full-length PARP1 activated by DNA (Table 1, Supplementary Fig. 2). HPF1 binds to full-length PARP1 with a $K_d$ of ~2.8 $\mu$M, slightly weaker than its binding affinity for CAT ΔHD, which is probably related to transient folding of the HD domain in solution.

**The crystal structure of human HPF1/PARP1-CAT ΔHD complex**. In order to understand the assembly and function of the HPF1/PARP1 complex, we determined the crystal structure of the human HPF1/PARP1-CAT ΔHD complex bound by benzamide (introduced when expressing the protein) and refined it to 1.98 Å (Fig. 2a, Supplementary Fig. 4 and Supplementary Table 1).

The high-resolution structure of the HPF1/PARP1-CAT ΔHD complex revealed an extensive interface between the two proteins that envelopes the region of the PARP1 active site (see interface I in Fig. 2a). The hetero-dimer binding interface was confirmed through extensive mutagenesis, combined with ITC and ADP-ribosylation assays. It has been well established that HPF1 binding to PARP1 restricts hyper-automodification of PARP1[23,26]. Indeed, hyper-automodification of PARP1 was clearly visible on SDS-PAGE as smeared bands above the unmodified PARP1 band, while wild-type HPF1 binding abolished this effect (Fig. 2b, see lanes 3 and 4). On the designated hetero-dimer interface, Phe268, Phe280, Asp283, Cys285, and Lys307 of HPF1 were found to directly interact with PARP1 residues (Fig. 2a, top-right and bottom-right insets), and these residues are conserved in HPF1 orthologs from different species (Supplementary Fig. 5). Indeed, mutating most of these HPF1 residues (F268S, F280A, D283H, C285H, and K307S) significantly restored hyper-automodification of PARP1 (Fig. 2b, see lanes 9, 11, 12, 14 and 15), indicating a loss

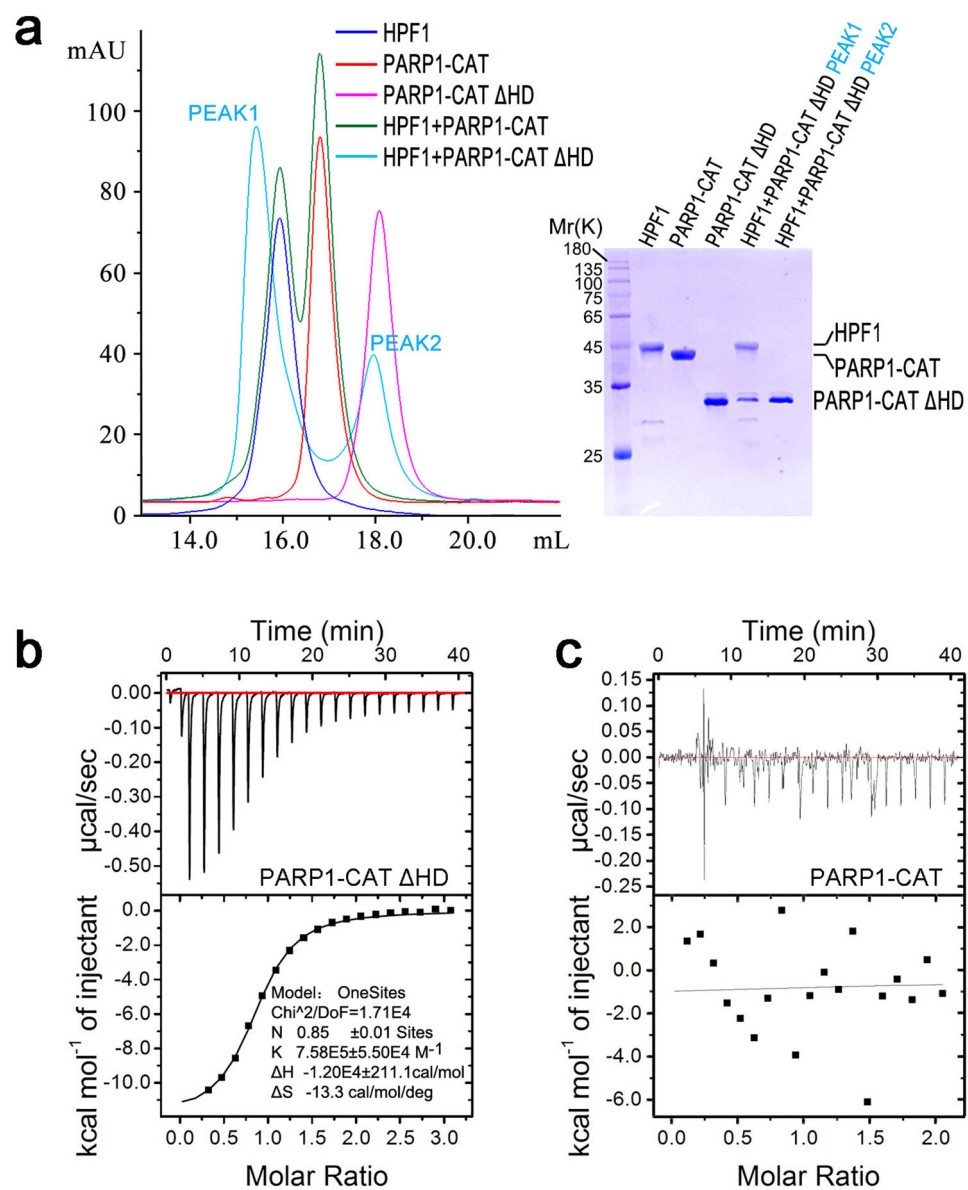

**Fig. 1 HPF1 binds to PARP1-CAT ΔHD. a** The SEC assay indicated that HPF1 binds to PARP1-CAT ΔHD, but not full-length PARP1-CAT. **b** Typical ITC data where human HPF1 was titrated into a solution of PARP-1 CAT ΔHD. In this titration, the association constant ($K$) of HPF1/PARP-1 CAT ΔHD was estimated to be 7.58E5 ± 5.50E4 $M^{-1}$, equivalent to a dissociation constant ($K_d$) of ~1.3 µM. After replicated measurements, the final $K_d$ value was determined to be 1.5 ± 0.2 µM (Table 1, Supplementary Fig. 2). **c** Typical ITC data where human HPF1 was titrated into a solution of full-length PARP-1 CAT. No binding was detected in this assay.

of binding between the HPF1 mutants and PARP1. ITC assays further confirmed that these mutations dramatically reduced the binding affinity between HPF1 and PARP1-CAT ΔHD (Table 1, Supplementary Fig. 2 and 3). Importantly, HPF1 F268S and D283H mutations completely abolished the interaction between these two proteins, and may be used in subsequent functional studies on the HPF1/PARP1 system. Interestingly, although PARP1, PARP2, and PARP3 share high amino acid sequence homology in the CAT domain (25% of the residues are identical and the other 19% are strongly similar), PARP3 lacks the key residues corresponding to His826, Leu985, Ser1012, Leu1013, and Trp1014 in PARP1. According to the HPF1/PARP1-CAT ΔHD complex structure and the mutagenesis data, these residues are important for the HPF1-PARP interaction. By contrast, these key residues are largely conserved between PARP1 and PARP2

(Fig. 2c, Supplementary Fig. 6). This explains why HPF1 can bind to PARP1 and PARP2, but not to PARP3[23].

In examining the structure, we noted another interface between a different HPF1 and PARP1 molecules in the crystal lattice (see interface II in Fig. 2a, top-left inset). However, mutating the interacting residues of HPF1 on this interface (Glu138, Phe139, and Lys216) did not interfere with PARP1 enzymatic activity or binding (Fig. 2b, see lanes 5–7; Table 1, Supplementary Figs. 2 and 3), and these residues are not conserved in HPF1 orthologs (Supplementary Fig. 5). Thus, this contact does not appear to be relevant for the catalytic function of the HPF1/PARP1 complex.

We also determined structures of human and mouse HPF1 alone, and observed that the conformation does not change upon binding to PARP1 (see Supplementary Fig. 7). Notably, analysis of the surface electrostatic potential showed that positively and

**Table 1 Dissociation constant ($K_d$) of HPF1 mutants binding to PARP1 CAT ΔHD and their effects on ADP-ribosylation of full-length PARP1 (automodification), histone H3, and HPF1.**

| Location of HPF1 mutants | $K_d$ (μM)[a] | ADP-ribosylation[b] | | |
|---|---|---|---|---|
| | | PARP1 | H3 | HPF1 |
| Control | | | | |
| Wild-type | 1.5 ± 0.2 | − | ++ | − |
| | 2.8 ± 0.3[c] | | | |
| Interface I[d] | | | | |
| F268S | N.D. | +++ | ++ | − |
| F280A | 6.0 ± 0.8 | +++ | ++ | − |
| D283H | N.D. | +++ | + | − |
| C285H | 28.7 ± 1.6 | +++ | ++ | − |
| K307S | 9.3 ± 1.2 | ++ | ++ | − |
| Interface II[d] | | | | |
| E138K | 1.7 ± 0.1 | − | ++ | − |
| F139S | 2.4 ± 0.1 | − | ++ | − |
| K216E | 1.4 ± 0.1 | − | ++ | − |
| Active center | | | | |
| R239A | 2.4 ± 0.1 | +++ | + | +++ |
| E284A | 0.3 ± 0.0 | + | − | + |

[a]$K_d$ data were determined in isothermal titration calorimetric (ITC) assays. The measurements were conducted in duplicates and the standard error values were listed. See Supplementary Fig. 2 and 3 for raw data.
[b]The assessment was made based on the ADP-ribosylation assays (see Fig. 2b). Number of "+" indicates the relative intensity of observed ADP-ribosylation.
[c]Titration of HPF1 to full-length PARP1 in the presence of DNA. Other $K_d$ values listed in this table are for titration of HPF1 to PARP1-CAT ΔHD.
[d]These interfaces were observed in the HPF1/PARP1-CAT ΔHD complex structure, indicating the two candidate interaction modes between HPF1 and PARP1.

negatively charged areas are evenly distributed on the surface of HPF1, except for part of helices α9/α10 and the linker loop connecting helices α6 and α7, where the protein surface is predominantly negatively charged (Supplementary Fig. 7B).

**HPF1 binding remodels the PARP1 active site for histone serine ADP-ribosylation.** Remarkably, our functional analysis of the structure shows that HPF1 and PARP1 form a composite active site, with HPF1 contributing key active-site residues. We first analyzed the surface electrostatic potential of HPF1, PARP1-CAT ΔHD, and the complex structures (Fig. 3a). Although PARP1 has been shown to play a key role in DDR, involving the modification of histones[16,28], the surface of the ART subdomain (especially the active site) is positively charged, which is clearly incompatible with the positively charged nucleosome (Supplementary Fig. 8). Interestingly, the binding of HPF1 to PARP1-CAT ΔHD dramatically changes the situation. As mentioned above, the surface covering helices α9/α10 and the loop connecting helices α6 and α7 of HPF1 is negatively charged (Supplementary Fig. 7). Many solvent-exposed acidic amino acid residues are located in this region, including Asp235, Glu240, Glu243, Asp245, Glu273, Asp283, Glu284, Asp286, and Glu292, while Arg239 is the only basic residue in this region (Fig. 3a). Notably, Arg239 and most of the acidic residues are conserved in HPF1 orthologs from different species (Supplementary Fig. 5). When binding to PARP1, this negatively charged region of HPF1 merges with the active site of PARP1 and changes its surface potential, creating an overall negatively charged joint active site that could potentially enable access to the positively charged histones (Fig. 3a). The surface of the HD subdomain facing the

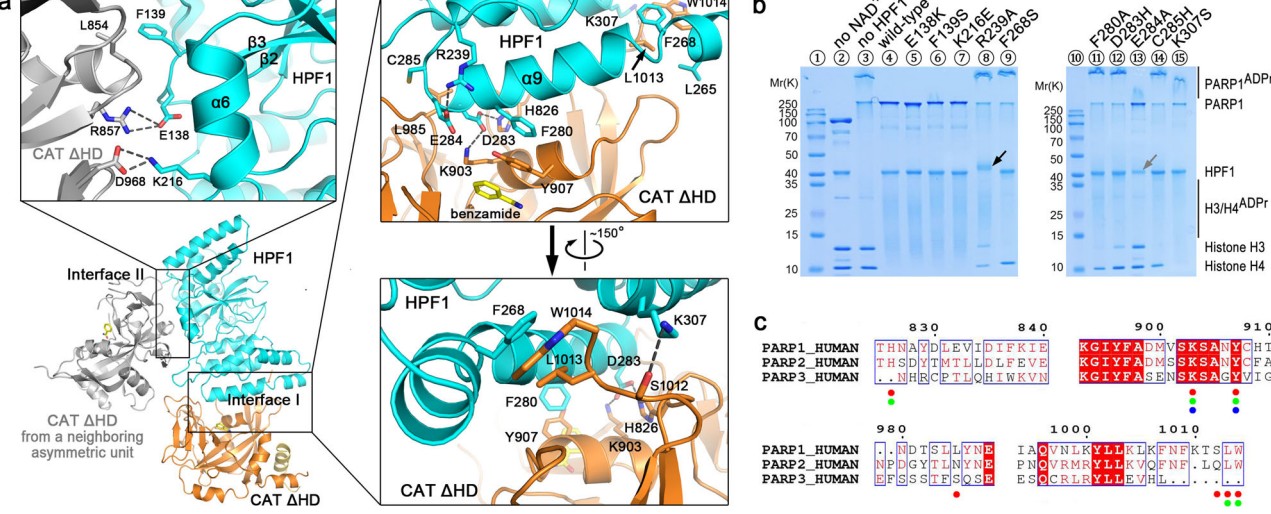

**Fig. 2 Crystal structure of human HPF1/PARP1-CAT ΔHD and the binding mode between these two proteins. a** Overall structure and the interaction details between human HPF1 and PARP1-CAT ΔHD. The overall structure of human HPF1/PARP1-CAT ΔHD is shown as cartoons. HPF1 and PARP1-CAT ΔHD are colored in cyan and orange, respectively. The gray-colored molecule on the left side indicates another PARP1-CAT ΔHD molecule from a neighboring crystallographic asymmetric unit that also contacts the HPF1 molecule, shown here due to close packing of molecules in the crystal. Therefore, the complex crystal structure indicated two possible interaction modes between HPF1 and PARP1-CAT ΔHD. The three insets show the key residues on the two interfaces that mediate the interactions between the HPF1 and PARP1 in the crystal. The dashed lines indicate polar interactions between the key residues. **b** Mutagenesis/ADP ribosylation studies to verify which key residues indicated in the crystal structure mediate HPF1 and PARP1 binding in solution. Hyper-automodification of PARP1 (shown by the smeared PARP1^ADPr bands on SDS-PAGE) was thought to indicate a loss of the interaction between HPF1 mutants and PARP1. Surprisingly, we noted that HPF1 R239A, but not any other HPF1 mutants, was robustly ADP-ribosylated in the assay (black arrow, also see Fig. 4c), while E284A seemed to be mildly ADP-ribosylated. The HPF1 E284A band became weaker, indicating a loss of unmodified protein in the assay (gray arrow). **c** Amino acid sequence alignment of human PARP1/2/3. The verified key residues that mediate HPF1 binding in PARP1 in solution are indicated by red dots. The corresponding residues in PARP2 and PARP3, if conserved in PARP1, are indicated by green and blue dots, respectively.

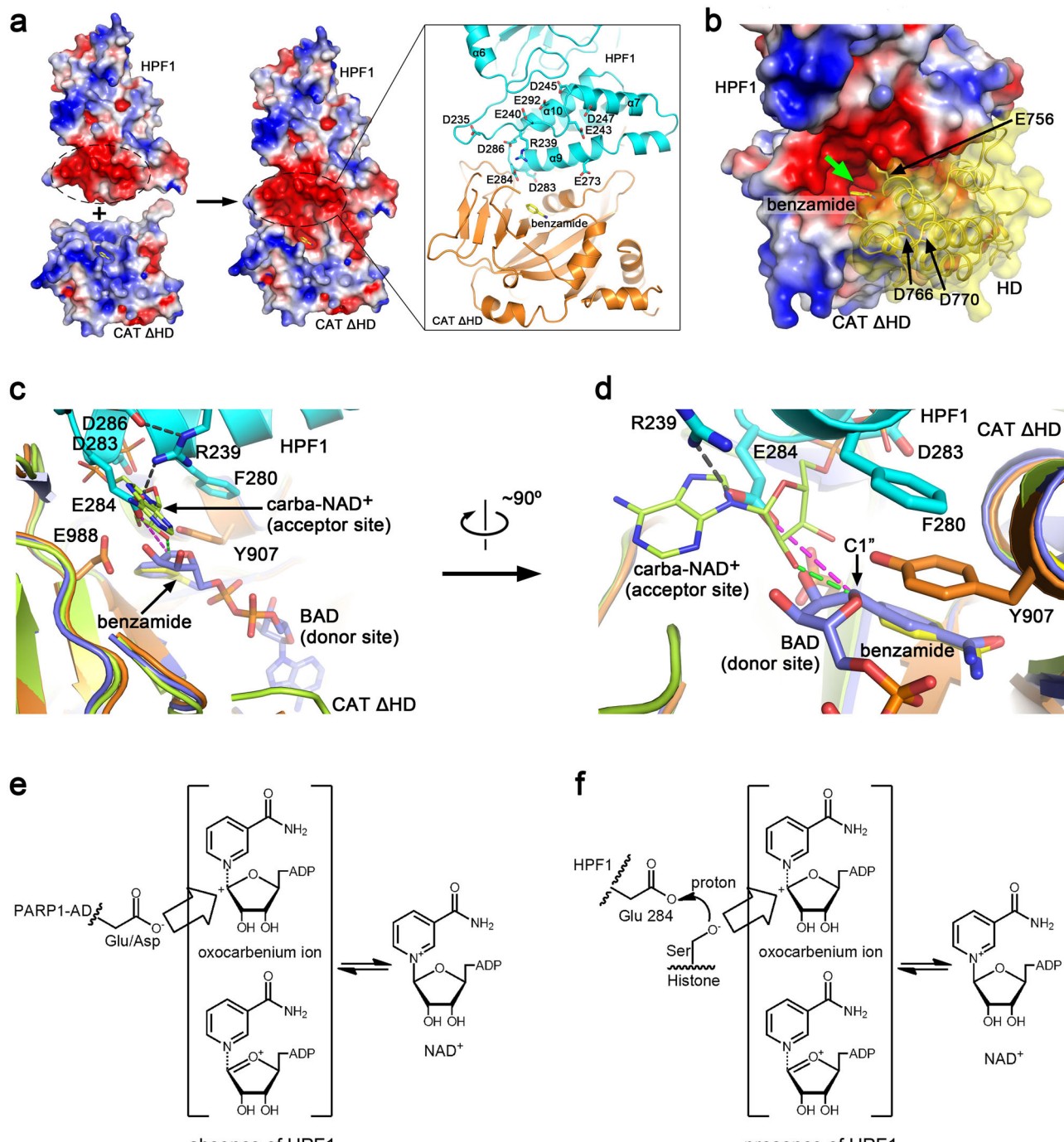

**Fig. 3 Structural basis of the mechanism through which HPF1 modulates PAPR1 activity. a** Surface electrostatic potential of human HPF1, PAPR1-CAT ΔHD and the hetero-dimer. The negatively and positively charged regions are colored red and blue, respectively. A strongly negatively charged region in HPF1 merges with the active site of PARP1-ART in the complex, creating a joint negatively charged active site. The inset on the right shows the acidic and basic residues in this negatively charged region of HPF1. **b** A close view of the remodeled active site and the HD domain if in the folded state. The HD domain is shown as cartoons covered by transparent surface based on a superimposition of the ART domain from the HPF1/PARP1-CAT ΔHD structure and the ART domain from the DNA-bound PARP1 crystal structure (PDB 4DQY). The green arrow indicates the entrance to the tunnel leading to the remodeled active center. **c**, **d** Superimposition of our HPF1/PAPR1-CAT ΔHD complex structure (cyan and orange) with the previously reported PARP1-ART/BAD complex structure (PDB 6BHV, light-blue)[22] and the PARP1-CAT/carba-NAD+ complex structure (PDB 1A26, lime)[29]. Two different views of the active cite are shown. BAD (light-blue sticks) is an NAD+ analog that mimics the ADP-ribosylation donor. The ADP moiety of carba-NAD+, shown as lime sticks in this figure, has been proposed to represent the ADP-ribosylation acceptor for ADPr chain elongation/branching. **e** A proposed catalytic mechanism of PARP1 automodification in the absence of HPF1. Without HPF1, the acidic residues in automodification domain (AD) of PARP1 can access the active center to get ADP-ribosylated. **f** Proposed catalytic mechanism of histone serine ADP-ribosylation in the presence of HPF1. Upon HPF1 binding, HPF1 Glu284 carboxyl is positioned approximately 4.6 Å (see the purple dashed lines in **c** and **d**) away from the donor NAD+ ribose (represented by BAD). This distance is too great for Glu284 itself to accept the ADP-ribose, but could catalyze ADP-ribosylation of a serine by promoting deprotonation of its sidechain hydroxyl.

active site is also negatively charged due to three acidic residues (Asp756, Asp766, and Asp770) and lacks any basic residues, resulting in the formation of a negatively charged narrow tunnel leading to the joint active site (Fig. 3b). However, since the HD subdomain is believed to undergo partial unfolding upon PARP1 activation[19], it is unclear if this tunnel is persistently maintained. We speculated that if the HD subdomain does not completely unfold, the negatively charged tunnel likely undergoes some conformational changes, but may still be maintained and is presumably most accessible for positively charged extended peptide substrates (e.g., the N-terminal tail of histone H3), while folded and/or negatively charged proteins may not readily access this joint active site for ADP-ribosylation.

A comparison of our HPF1/PARP1-CAT ΔHD complex crystal structure with the previously reported PARP1-CAT ΔHD/BAD complex crystal structure (PDB 6BHV)[22] and PARP1-CAT/ carba-NAD$^+$ complex crystal structure (PDB 1A26)[29] provide interesting insights (Fig. 3c and d). The benzamide molecule in our structure clearly overlaps with the benzamide moiety of BAD (an NAD$^+$ analog mimicking the ADP-ribosyl donor in the reaction). The HPF1 Glu284 side-chain carboxyl (positioned/ locked by the Arg239-Glu284 salt-bridge) is located next to the ribose of the ADP moiety of carba-NAD$^+$, and is about 1.9 Å further away than the latter from the C1" atom of the nicotinamide ribose moiety of BAD/NAD$^+$ (see purple versus green dashed lines in Fig. 3d). Since the 2′-hydroxyl of the ADP moiety of carba-NAD$^+$ is believed to attack the C1" atom of the nicotinamide ribose during poly(ADP-ribose) chain elongation[29], the distance between them (the green dashed line in Fig. 3d) indicate a suitable distance for the attacking. Therefore, the much longer distance of HPF1 Glu284 carboxyl from C1" atom of the nicotinamide ribose (the purple dashed line in Fig. 3d) would strongly suggest that Glu284 acts as a catalyst for ADP-ribose transfer to a third party (e.g. a serine residue), rather than as a direct acceptor of ADP-ribose. According to the current understanding of ADP-ribosylation[30], the positively charged nicotinamide nitrogen of NAD$^+$ tends to abstract an electron from the connecting ribose C1" carbon, rendering the latter into an oxocarbenium ion that can then be attacked by a nucleophile, such as a negatively charged/deprotonated amino acid side-chain. Since acidic amino acids such as glutamate and aspartate may spontaneously lose a proton to become negatively charged at physiological pH, these residues can be readily ADP-ribosylated when approaching the active center of PARP1 in the absence of HPF1 (Fig. 3e). In the case of the HPF1/PARP1 complex, however, the carboxyl of HPF1 Glu284 is too far away (~4.6 Å) from the NAD$^+$ to attack the ribose C1" carbon (Fig. 3d, purple dashed line). It therefore seems unlikely that Glu284 itself can become ADP-ribosylated. Regarding serine ADP-ribosylation, the electroneutral serine residue must be activated, i.e. deprotonated, by a third party to enable nucleophilic attack. The negatively charged Glu284 of HPF1 or Glu988 of PARP1 near the active site may facilitate this process (Fig. 3c). However, PARP1 is unable to ADP-ribosylate serine without HPF1[24], and the PAPR1 E988Q mutation does not abolish serine ADP-ribosylation by the PARP1/HPF1 complex[31]. These observations indicate that PARP1 Glu988 is not the key residue to facilitate serine ADP-ribosylation. Upon mutating HPF1 Glu284 to alanine, we observed that HPF1 E284A abolished the ADP-ribosylation of histones (Fig. 2b, lane 13). This observation indicates that Glu284 is the key catalytic residue for serine ADP-ribosylation following the binding of HPF1 to PARP1, which leads to the formation of a joint active site. The same observation and conclusion were reported by Suskiewicz et al. in their structural and functional studies of the HPF1/PARP2-CAT ΔHD complex[27].

**Why does HPF1 binding hinder PARP1 automodification and poly-ADP-ribosylation?** Based on the verified interaction mode of HPF1 with PARP1-CAT ΔHD (corresponding to interface I shown in Fig. 2a), an overall model of HPF1 binding to DNA damage-activated PARP1 was constructed by overlapping our HPF1/PARP1-CAT ΔHD complex structure with the PARP1/ DNA complex structure (PDB 4DQY) (Fig. 4a)[18]. Binding of HPF1 to PARP1 is compatible with other domains of the activated enzyme. This model illustrates why HPF1 binding restricts the automodification of PARP1. According to a previous model proposed by Langelier et al., the AD/BRCT domain carrying most of the modified sites is located close to the CAT domain, presumably for automodification[18] (Supplementary Fig. 1), and is roughly in the same location where HPF1 resides (compare Fig. 4a to Supplementary Fig. 1). We therefore speculate that HPF1 binding would dislodge AD/BRCT from this location, hence limiting the automodification of PARP1, at least in the folded BRCT domain.

The HPF1/PARP1-CAT ΔHD complex structure also suggests that HPF1 not only restricts hyper-automodification of PARP1, but may switch the modification from poly-ADP-ribosylation to mono-ADP-ribosylation. An early study by Ruf et al. observed the binding of the ADP moiety of carba-NAD$^+$ in the active site (PDB 1A26), and this ADP moiety was believed to represent the acceptor for the next ADP-ribose during poly(ADP-ribose) chain elongation and branching[29]. Compared to the HPF1/PARP1-CAT ΔHD complex structure reported here, we noted that the ADP (i.e. the acceptor for the next ADP-ribose in poly-ADP-ribosylation) binding site is partly occupied by HPF1 residues Asp283 and Glu284 (Fig. 3c and d), implying that HPF1 binding is likely to abolish poly(ADP-ribose) chain elongation and branching. However, poly-ADP-ribosylation may still take place, likely with lower efficiency, due to the transient dissociation of HPF1 from PARP1 as is indicated by the long smeared bands observed in the in vitro ADP-ribosylation assay (Fig. 2b, see histone H3/H4$^{ADPr}$).

**The unique role of HPF1 Arg239.** Arg239 is highly conserved in HPF1 orthologs (Supplementary Fig. 5), and is thought to be important for the interaction between HPF1 and PARPs. Mutating this residue to alanine (R239A) restores automodification of PARP1 (Fig. 2b, lane 8), which was previously interpreted as loss of binding of HPF1 to PARP1[23].

However, in the human HPF1/PARP1-CAT ΔHD (reported here) and HPF1/PARP2-CAT ΔHD (reported by Suskiewicz et al.[27]) complex structures, HPF1 Arg239 was found to only interact with Glu284 and Asp286, but not with any residue from PARP1/2 (Fig. 4b). Furthermore, our ITC assays confirmed that R239A only mildly reduced the binding affinity between HPF1 and PARP1 (Table 1). In the ADP-ribosylation assays, the band of HPF1 R239A, but not of any other HPF1 mutants, clearly shifted and smeared on the SDS-PAGE (Fig. 2b, lane 8, black arrow), likely indicating that HPF1 R239A itself was ADP-ribosylated. Analysis of the shifted HPF1 R239A band by mass spectrometry confirmed that HPF1 R239A was ADP-ribosylated on Asp235 and Glu240, two acidic residues flanking Arg239 in the long loop (residues 217–244) connecting helices α6 and α7 (Fig. 4b and c). Long loop regions are often difficult to resolve in crystal structures due to their intrinsic flexibility, but in HPF1, the loop connecting helices α6/α7 is ordered and well resolved. As is shown in Fig. 4b, the C-terminal half of this long loop is stabilized by two groups of interactions: one is the salt-bridge network including Arg239-Asp286 and Arg239-Glu284, which also acts to position the catalytically important Glu284 side-chain; the other is the hydrophobic interaction between Tyr238 phenyl and

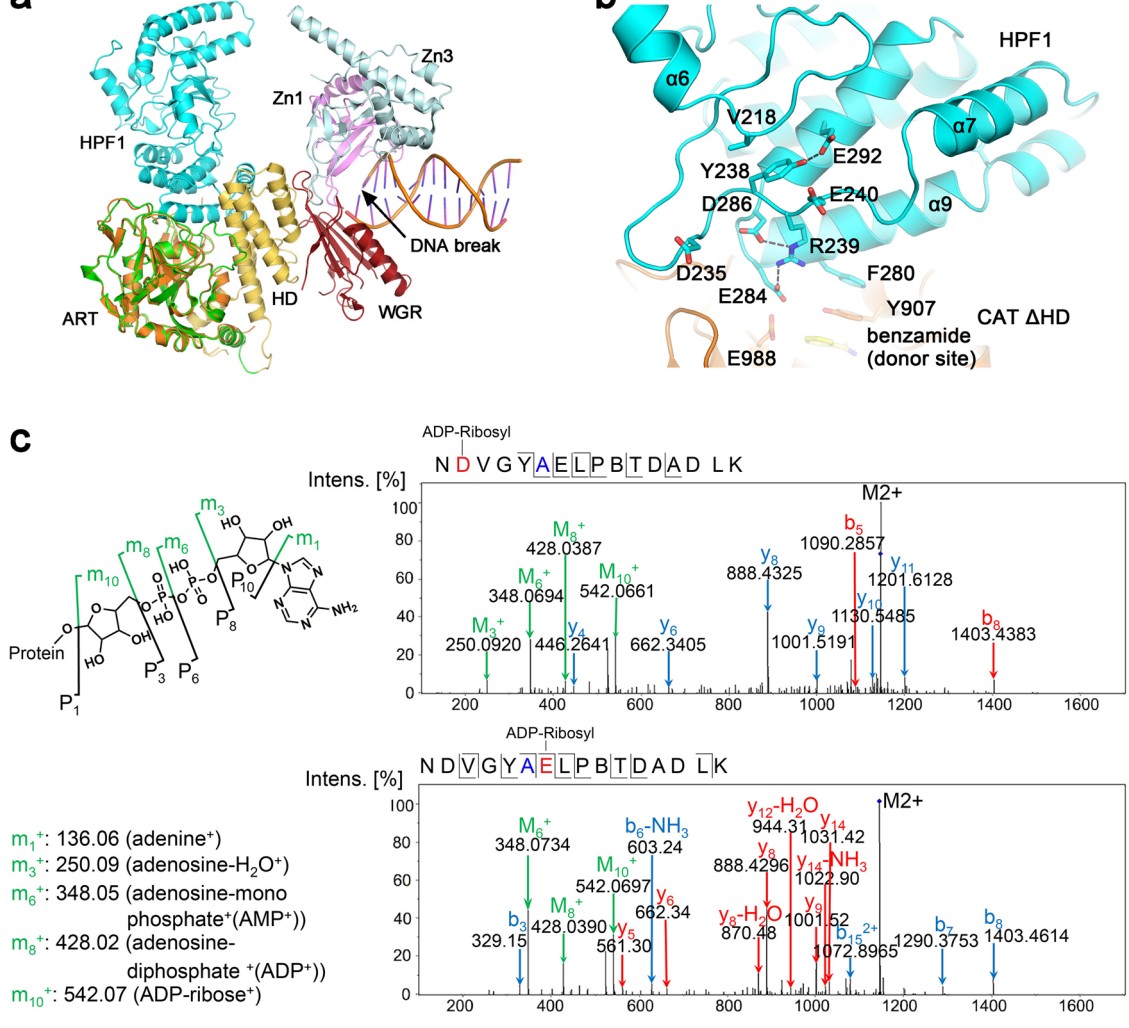

**Fig. 4 Depicting the function of HPF1 Arg239. a** Comparison of the HPF1/PAPR1-CAT ΔHD complex structure and the DNA-bound PARP1 crystal structure (PDB 4DQY)[18] by superimposing the ART domains from both (orange—ART determined in this report; green—ART of PDB 4DQY). HPF1 binding to PARP1 would presumably dislodge PARP1 automodification domain (AD) that was previously proposed to be located roughly in the same position where HPF1 binds (see Supplementary Fig. 1), thus prevents automodification of PARP1. **b** A close view of the long α6-α7 loop region structure and its position relative to the active center of the HPF1/PARP1 complex. **c** Mass Spectrometry analysis revealed that HPF1 R239A mutation resulted in ADP-ribosylation on Asp235 and Glu240, a unique phenomenon only seen in this mutant (see Fig. 2B, black arrow). The mutated residue 239 is shown in blue in the sequence, while the modified residues Asp235 and Glu240 are shown in red.

Val218 side-chain, as well as hydrogen bonding between Tyr238 hydroxyl and Glu292 carboxyl. It can be inferred that when Arg239 was mutated to alanine and lost the ability to interact with Glu284 and Asp286, the loop became so flexible that Asp235 and Glu240 gained the freedom to attack NAD$^+$ in the active center and became ADP-ribosylated. Interestingly, although HPF1 R239A still bound PARP1 pretty well, it also restored the automodification of PARP1 to a great extent (Fig. 2b, lane 8). The PARP1 automodification domain (AD) contains the folded subdomain BRCT (residues 380–480) and an unstructured automodification peptide (residues 481–530). In a previously reported in vitro ADP-ribosylation assay, the serine residues in the Lys-Ser motif within the automodification peptide (Ser499, Ser507, and Ser519) were ADP-ribosylated in the presence of HPF1[24]. This ADP-ribosylation was not observed in the BRCT subdomain (Lys467-Ser468), indicating that the Lys-Ser motif in a flexible peptide, but not in a folded domain, may access the remodeled HPF1/PARP1 active center under certain conditions. In our assay, the robust binding of wild-type HPF1 or the F139S

mutant (HPF F139S mutant bound to PARP1-CAT ΔHD with similar affinity as the R239A mutant, see Table 1) limited the automodification of PARP1, while R239A, although still bound to PARP1, restored the automodification, likely due to loss of rigidity of the loop stabilized by Arg239 in this mutant. Taken together, our data indicate that the rigidity of the loop region strengthened by Arg239-Glu284/Arg239-Asp286 interactions is essential for limiting PARP1 automodification in the unstructured automodification peptide region.

In the in vitro ADP-ribosylation assay, we also noted that the HPF1 E284A band became thinner (Fig. 2b, lane 13, gray arrow), indicating the loss of a small amount of the unmodified protein (or ADP-ribosylation of a small portion of the protein). However, the modification of this mutant HPF1 protein must be minor, since no shifted band was clearly visible on the gel. We speculate that in this mutant, after losing the Arg239-Glu284 salt-bridge but still maintaining the Arg239-Asp286 salt-bridge, the loop region gained some flexibility but the acidic residues (e.g. Asp235 and Glu240) did not gain enough freedom to attack NAD$^+$ and

become ADP-ribosylated. Accordingly, HPF1 E284A mildly restored PARP1 automodification (Fig. 2b, lane 13) though this mutant enhanced the binding between HPF1 and PARP1, which again supported the notion that the rigidity of this loop region strengthened by Arg239-Glu284/Arg239-Asp286 interactions is essential for limiting PARP1 automodification.

**Indirect contribution of Arg239 to HPF1/PARP1 binding.** The discussed observations do not explain why R239A mutation weakens the interaction between HPF1 and PARP1, since no residue in the loop region (including Arg239 itself) directly interacts with PARP1. We noted that the HPF1 E284A mutant showed a unique behavior in our ITC assays. It is the only mutation that enhanced the binding between HPF1 and PARP1 (Table 1). Taking into account the negative electrostatic properties of the joint active site discussed above, we propose that the presence of the negatively charged Glu284 in the active center is not favorable for HPF1/PARP1 binding, while the salt-bridge with the positively charged Arg239 side-chain may act to neutralize the negative charge on the Glu284 carboxyl, thus facilitating the binding. Mutating Arg239 to directly break the Arg239-Glu284 interaction, or mutating Tyr238 to destroy the local conformation supporting the Arg239-Glu284 interaction, would restore the negative charge of Glu284, thus weakening HPF1/PARP1 binding. This explains why the HPF1 R239A (and/ or Y238A[23]) mutation weakens the binding between the two proteins. Although mutating Arg239 (and/or Tyr238) appears to weaken HPF1/PARP1 binding, it may not be the right choice for setting the non-binding control in functional studies, as the mutation(s) do not fully abolish binding and may have other structural and functional implications.

Taken together, our study indicates that Arg239 is a key residue acting to 1) position Glu284 at the right location for catalyzing serine ADP-ribosylation; 2) stabilize the conformation of the long loop region across Arg239 to limit automodification of PARP1; and 3) facilitate HPF1/PARP1 binding by neutralizing the negative charge of the Glu284 side-chain.

## Discussion

Structures of the sea anemone and human HPF1 (both at 2.09 Å resolution) and human HPF1/PARP2-CAT ΔHD complex (at 2.96 Å resolution) determined by Suskiewicz et al. provided a first glance of the interaction between HPF1 and a PARP family member, and provided insights into how this interaction completes the PARP2 active site and promotes ADP-ribosylation of histone serines[27]. This mechanism was thought to also apply to PARP1, based on NMR and functional studies[27]. Here, we independently determined the crystal structures of mouse and human HPF1 (at 1.71 Å and 1.57 Å resolution, respectively), as well as the human HPF1/PARP1-CAT ΔHD complex (at 1.98 Å resolution). This has provided direct structural insights into how HPF1 binds to PARP1, why HPF1 binds to PARP1/2 but not PARP3, and why this interaction limits the hyper-automodification of PARP1/2 and may also limit poly-ADP-ribosylation. The HPF1/PARP1-CAT ΔHD complex also hinted why HPF1 binding promotes histone peptide recognition and serine ADP-ribosylation. Since PARP1 and PARP2 share high amino acid sequence homology in the CAT domain (~46% identity), the structures reported by Suskiewicz et al. are similar to the structures in this work (Supplementary Fig. 9), and our structural and functional studies confirmed most of the conclusions made by Suskiewicz et al.

In addition to confirming the key findings of Suskiewicz et al., our work provides additional insights into the structure-function relationships of the HPF1/PARP1 complex. Since multiple contacting interfaces present in crystal structure due to protein molecules packing, caution must be taken to carefully verify which observed interface in the complex crystal structure represents the true functional dimer interface. In our study, this was done through extensive mutagenesis, combined with functional studies. When studying the function of HPF1 mutants to verify the true dimer interface, we conducted traditional ADP-ribosylation assays (Fig. 2b), which serve as an indirect qualitative method to determine if the HPF1/PARP1 complex is formed. We further confirmed the formation of the complex by measuring the $K_d$ values using ITC assays (Table 1). These data quantitatively and directly described the binding behavior between HPF1 and PARP1, and showed that only certain mutations (F268S and D283H) can completely abolish the binding. We suggest that subsequent functional studies, especially cell-based studies, utilize these mutations for the construction of non-binding controls.

In previous studies, residue Arg239 of HPF1 was thought to mediate the interaction between HPF1 and PARP1/2, but this idea is not supported by the structures. The same confusion also applies to Tyr238, a residue located inside the HPF1 molecule. To resolve this problem, we first asked if Arg239 interacts with some other PARP1 residue not present in our HPF1/PARP1-CAT ΔHD complex structure. Based on the structural model (Fig. 4a), the PARP1 HD subdomain seems to be the only domain that could directly interact with HPF1 Arg239, and Asp756 in the HD subdomain seems to be the residue most likely to form a salt-bridge with HPF1 Arg239. However, it has been shown that the HD subdomain hinders the binding of HPF1, and our muta-genesis studies on PARP1 D756A showed that this mutation had no effect on HPF1/PARP1 binding (Supplementary Fig. 2). Furthermore, the ITC data confirmed that HPF1 R239A still binds to PARP1, and mass spectrometry confirmed that this mutant protein was ADP-ribosylated in the ADP-ribosylation assay. Taking into account the observation that HPF1 R239A significantly restored PARP1 automodification, we propose that Arg239 (and presumably also Tyr238) acts at the central stage to stabilize the local conformation of the α6-to-α7 loop region by interlocking with Glu284 and Asp286 (and by intramolecular interactions with Tyr238), which is essential for limiting PARP1 automodification, but dispensable for histone serine modification. Surprisingly, the HPF1 E284A mutant binds to PARP1-CAT ΔHD with about 3 times higher affinity than wild-type HPF1, which suggests that the negative charge of Glu284 is not favored for HPF1/PARP1 binding and that neutralizing the negative charge of Glu284 by Arg239 facilitates binding. In accordance with the observation that HPF1 R239A loses the Arg239-Glu284 and Arg239-Asp286 salt-bridges, the HPF1 E284A mutant, losing only the Arg239-Glu284 salt-bridge, only partly restored PARP1 automodification (Fig. 2b). These data further support our conclusion that Arg239 functions to stabilize the local conformation and limit PARP1 automodification. Since the function of Arg239 is more complex than previously thought, utilizing this mutation as a control to abolish HPF1 and PARP1/2 binding in cell-based studies may result in misleading observations.

Our work provides an in-depth examination of the complex interactions between HPF1 and PARP1, the most important member of the PARP family. The high-resolution structures presented here, along with the extensive mutagenesis studies and quantitative functional data, complement previous studies done by Suskiewicz et al. and address unresolved questions. Taken together, these findings can have important implications for the design of drugs targeting PARP1/2. Since histone serine ADP-ribosylation is a key step in DDR[8,9,13,16,26], and turning off PARP1/2 activity to inhibit the DNA damage response in tumor cells is a promising approach for cancer treatment, it is worthwhile to develop inhibitors targeting HPF1-dependent serine

ADP-ribosylation, i.e. to consider the remodeled active site structure, instead of the open active site of PARP1/2 alone.

## Materials and methods

**Cloning, expression, and purification of HPF1 and PARP1.** Full-length (FL) human and mouse HPF1 cDNA was synthesized by Tsingke Bio-tech® (Beijing) and full-length PARP1 plasmid was a kind gift from Professor Jia-Dong Wang's laboratory in Peking University Health Science Center. The full-length and truncations of HPF1 and PARP1 (FL, CAT 661-1014 AA, CAT△HD 661-678-GSGSGSGG-787–1014 AA) were all sub-cloned into pET-28a vector. An N-terminal 6×His-tag followed by a tobacco etch virus (TEV) protease cleavage site (ENLYFQG) was added to the N-terminus of the constructs to facilitate protein purification. Mutations were introduced using the QuickChange Site-Directed Mutagenesis Kit (Strategene). All constructs were verified by sequencing by Ruibiotech® (Beijing).

The HPF1 and PARP1 proteins were expressed in *Escherichia coli* Rosetta (DE3). Once expression of PARP1-CAT ΔHD commenced, the inhibitor benzamide was added to the media at 10 mM final concentration to reduce toxicity to host cells, as previously described.

The same protocol was followed to purify HPF1 and PARP1 constructs. In brief, harvested bacteria cells were lysed by sonication in buffer A (20 mM Tris-pH 8.0, 500 mM NaCl, 1 mM TCEP, 5% glycerol) and the lysate was cleared by centrifugation (18,000 × g at 4 °C for 20 min). The supernatant was loaded onto a Ni²⁺ affinity column (GE Healthcare), washed sufficiently with buffer A supplemented with 35 mM imidazole, and then eluted using buffer A supplemented with 400 mM imidazole. The His-tag was then removed by incubating the eluent with His-tagged TEV protease at room temperature for 2 h. The uncleaved target proteins and His-tagged TEV protease were subsequently removed by passing the sample through the Ni²⁺ affinity column for the second time. The flow-through was collected, concentrated, and applied to gel-filtration for the final round of purification using a Superdex 200 Increase column (GE Healthcare) in buffer B (20 mM Tris-pH 8.0, 150 mM NaCl, 1 mM TCEP, 1% glycerol). When purifying full-length PARP1, an ion-exchange step using anion exchange column was added after the first round of Ni²⁺ affinity chromatography. The final purified protein samples were concentrated to a final concentration of 30 mg/mL. Aliquots were made and flash-frozen in liquid nitrogen and stored at −80 °C.

**Crystallization, data collection and structure determination.** All HPF1 crystals used in this study were made by hanging drop vapor diffusion. The crystallization reservoir solution for human HPF1 34-346 was 0.1 M Tris-pH 8.1, 0.2 M potassium acetate, 20% w/v Polyethylene glycol (PEG) 3350; while the crystallization reservoir solution for mouse HPF1 (mouse HPF1) 26-346 was 0.2 M Potassium citrate tri-basic monohydrate, 20% w/v PEG 3350, pH7.9. Heavy atom doped crystals of human HPF1 34-346 were made by soaking the native crystals in cryo-protectant (crystallization reservoir solution supplemented with 20% ethylene glycol) containing 1 mM ethylmercuric chloride (C₂H₅HgCl) for 10 to 30 min.

To obtain the HPF1/PARP1-CAT △HD complex crystals, human HPF1 26-346 and PARP1-CAT △HD were mixed at a molar ratio of 1:1 and adjusted to 0.5 mM prior to crystallization. The HPF1/PARP1-CAT △HD complex crystals were obtained by sitting drop vapor diffusion. The reservoir solution was 0.01 M Tris-pH 7.0, 0.2 M magnesium formate dehydrate, 20% w/v PEG 3350. The cryo-protectant for the complex crystals was the crystallization reservoir solution supplemented with 25% PEG 400.

All crystals were soaked in their corresponding cryo-protectants and then flash-frozen in liquid nitrogen. X-ray diffraction datasets were obtained at Shanghai Synchrotron Radiation Facility (SSRF) BL18U1 or BL19U1 at 100 K, and the data were processed with HKL3000[32]. The human HPF1 crystal structure was solved by single-wavelength anomalous diffraction (SAD) using the ethylmercuric chloride derivative dataset and the SHARP software package[33]. The structure model was then refined against the native human HPF1 dataset with the higher resolution. The mouse HPF1 and human HPF1/PARP1-CAT △HD complex crystal structures were solved by molecular replacement with Phaser[34], using human HPF1 structure and the previously reported PARP1-CAT △HD crystal structure (PDB 6BHV)[22] as the search models. Iterative cycles of manual refitting and crystallographic refinement were performed using COOT[35] and Phenix[36]. All structure figures were prepared by PyMOL program (Schrödinger LLC, http://www.pymol.org/). Statistics of diffraction data processing and structure refinement are shown in Supplementary Table 1.

The human HPF1 34-346, mouse HPF1 26-346 and human HPF1(26-346)/PARP1-CAT △HD crystal structures have been deposited in the Protein Data Bank (http://www.rcsb.org) with the accession IDs 6M3G, 6M3H and 6M3I, respectively.

**Size-exclusion chromatography.** 100 μL samples (HPF1, CAT, CAT△HD, HPF1/CAT mixture, HPF1/CAT△HD mixture) containing 50 μM of each protein were prepared and incubated on ice for at least 30 min. Samples were then put on a Superdex 200 Increase column (GE Healthcare) and eluted with a buffer consisting of 20 mM Tris-pH 8.0, 150 mM NaCl, 1 mM TCEP, 1 mM MgCl₂. Samples from peak fractions were resolved on SDS-PAGE gel and detected using Coomassie blue staining.

**Isothermal titration calorimetry (ITC) assay.** The ITC assays were carried out using a MicroCalorimeter ITC200 (Microcal LLC) at 25 °C in buffer composed of 20 mM Tris-pH 8.0, 150 mM NaCl, 1 mM TCEP, 1 mM MgCl₂. HPF1 wild-type and mutants were prepared as 0.3 or 0.4 mM solution and put in the syringe, while PARP1 CAT-ΔHD protein was prepared as 0.02 mM solutions in the calorimeter cell. When analyzing the full-length PARP1, protein of 0.02 mM concentration was pre-incubated with 0.06 mM DNA hairpin (prepared by annealing 5′-GCCTACC GGTTCGTCTGGTGACGAACCGGTAGGC-3′ DNA fragment) at 4 °C for at least 60 min. The titration data were processed with MicroCal Software.

**ADP-ribosylation assay.** This assay was conducted based on SDS-PAGE. The reaction consisted of 5 μM HPF1, 5 μM full-length PARP1, and 10 μM 15 bp DNA hairpin in a buffer composed of 20 mM Tris-pH 8.0, 150 mM NaCl, 1 mM MgCl₂. The modified substrate H3/H4 tetramer were used at 10 μM. NAD⁺ was added to the mixture to the final concentration of 5 mM to start the reaction. The reaction was carried out at 20 °C for 10 min and then terminated with SDS-PAGE loading buffer, which denatures proteins. The samples were then resolved on a 12 or 15% polyacrylamide gel. The DNA hairpin used in this study was prepared by first denaturing the synthesized oligonucleotides (5′-GCCTACCGGT TCGTCTGGTGACGAACCGGTAGGC-3′) at 95 °C for 5 min and then quickly cooling down on ice. Presence of the double-strand DNA was confirmed by agarose gel electrophoresis.

**Mass spectrometry assay**

*Sample preparation for LC-MS analysis.* Two samples were analyzed in the study: the PARP1 and HPF1 R239A mixture prepared in the ADP-ribosylation assay, and the pure HPF1 R239A as the control. The two samples were precipitated with 60 μL acetone at -20 °C for 4 h, and centrifuged at 16,000×g for 30 min at 4 °C. After removal of the supernatant, the precipitate was dried using a vacuum concentrator (Labconco, USA). The dried precipitate was resuspended in 40 μL 8 M Urea in 500 mM Tris-HCl buffer (pH 8.5), incubated with 20 mM (2-carboxyethyl) phosphine hydrochloride (TCEP) (500 mM in 100 mM Tris/HCl pH 8.5) at room temperature for 20 min, then alkylated with 40 mM IAA in the dark for 30 min. The mixture was diluted with 200 μL 100 mM Tris-HCl buffer (pH 8.5) to the final concentration of 1.3 M Urea. The two samples were divided into 4 equal parts, followed by digestion with trypsin, elastase, GluC or chymotrypsin protease at a 1/50 (w/w) protease to protein ratio, at 37 °C overnight. The digestion was quenched by the addition of formic acid at a final concentration of 5%. Samples were desalted using Monospin C18 columns (GL Science, Tokyo, Japan). The purified peptides were vacuum-centrifuged until dry and reconstituted in Milli-Q water with 0.1% (v/v) formic acid (FA). The hydrolysates of four proteases from two samples were mixed respectively for LC-MS analysis.

*Liquid chromatography.* We employed a nanoElute liquid chromatography system (Bruker Daltonics). Peptides (200 ng of digest) were separated over 240 min at a flow rate of 300 nL/min on a 25 cm×75 μm column with a laser-pulled electrospray emitter packed with 1.5 μM ReproSil-Pur 120 C18-AQ particles (Dr. Maisch). Mobile phases A and B were water and ACN with 0.1% (v/v) formic acid. The %B was linearly increased from 2 to 22% in 180 min, followed by an increase to 37% in 20 min and a further increase to 95% in 20 min, followed by a final step at 95% for 20 min.

*Mass spectrometry.* The two samples were analyzed on a hybrid TIMS quadrupole time-of-flight mass spectrometer (Bruker tims TOF Pro) via a Captive Spray nano-electrospray ion source. The mass spectrometer was operated in data-dependent mode. Accumulation and ramp time were set to 100 ms each, and recorded mass spectra in the range from m/z 100–1700 in positive electrospray mode. The ion mobility was scanned from 0.6 to 1.6 Vs/cm². The overall acquisition cycle of 1.16 s comprised one full TIMS-MS scan and 10 PASEF MS/MS scans.

*Data analysis.* Protein and ADP-Ribosyl modification analysis were performed with Thermo Proteome Discoverer 2.3 (Thermo Fisher, San Jose, CA) and were searched against a homemade database (only reviewed entries) which combined the human UNIPROT database (only reviewed entries) with two protein (PARP1 and HPF1) entries. A decoy database containing the reversed sequences of all the proteins were employed to estimate the FDR (set at 0.01) at both peptide and protein levels. The search space included all fully- and half-protease digested peptide candidates that fell within the mass tolerance window (MS scan, 20ppm; MS/MS scan, 0.1 Da). Carbamidomethylation (+57.02146 Da) of cysteine was considered as a static modification, while ADP-Ribosyl (+542.061 Da) on aspartic acid, glutamic acid, lysine, asparagine, serine, threonine were considered as variable modifications.

**Reporting summary.** Further information on research design is available in the Nature Research Reporting Summary linked to this article.

## Data availability

The coordinates of mouse HPF1, human HPF1 and human HPF1/PARP1-CAT ΔHD structures, and the corresponding diffraction data have been deposited in Protein Data

Bank (https://www.rcsb.org/) with the accession codes 6M3H, 6M3G and 6M3I, respectively. The raw data/image of SEC, SDS-PAGE and mass spectra are shown in the corresponding figures (Figs. 1, 2, and 4). The Raw data of ITC are shown in Supplementary Figs. 2 and 3. All other data are available from the corresponding author upon reasonable request.

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

## Acknowledgements

We thank the staff of APS beamline ID-19 and SSRF beamlines BL18U/BL19U1 for assistance in data collection. We thank Prof. Jia-Dong Wang for providing the PARP1 cDNA. We thank Drs. Michael J Eck, Oreste Segatto and Jianming Zhang for helpful discussions. This study was funded by the National Natural Science Foundation of China (No. 31270769), the National Basic Research Program of China (973 Program, No. 2012CB917202), and the Ministry of Education of China (NCET-12-0013).

## Author contributions

C.-H.Y. instructed the project. F.-H.S. and P.Z. designed and performed the crystallographic and functional experiments. L.-L.K. constructed several expression vectors used in this study. N. Z. and C.C.L.W. conducted the Mass Spectrometry studies. F.H.S., C.C. L.W. and C.H.Y. analyzed the data and wrote the manuscript.

## Competing interests

The authors declare no competing interests.
