## [Peer Review File · Nature Communications]

REVIEWERS' COMMENTS

Reviewer #1 (Remarks to the Author):

Sun et al present the crystal structures of HPF1 and the HPF1:PARP1 complex, and a brief biochemical characterization that strengthens the mode of interaction and PARP1 activation/re-direction of specificity.

It is problematic that the authors remain closely attached - in terms of study content, scientific hypotheses, and methodology - to a previous publication (Suskiewicz et al., *Nature* 579: 598-602) with very similar content and results. One exception are the new ITC data; unfortunately, the authors use ITC only to confirm what we already knew about the complex. Throughout the text the authors stress the small differences to the previous paper: The present study shows the PARP1 (not PARP2) complex with HPF1, and with better resolution than before. Nevertheless, given the high degree of homology between the two enzymes, especially within the active site; and given the extensive experimental verification carried out in the previous paper, it is clear that the present manuscript only adds a minor increment to our understanding of PARP1 activity during the DNA damage response. In fact, this appears to be the authors' own conclusion (see line 22-24).

It is unfortunate that the authors chose not to pursue any questions and concepts that were left unanswered by the previous study, including questions they mention in their text. For instance, how can the new findings inform drug discovery (mentioned in several instances, but not addressed by experiment)? Are there any PARP inhibitors that preferentially inhibit either the HPF1 complex or PARP1 alone (i.e., serine ADP-ribosylation vs carboxylic acid ADP-ribosylation)? Similarly, the authors might have chosen to examine the consequences of their and the Ahel group's findings in cells during DNA stress conditions.

In summary, I believe that although this study is timely, the results presented have limited novelty and originality.

Reviewer #2 (Remarks to the Author):

I thank the authors for their thorough response to my previous comments! I welcome the extensive analysis of mutant variants and the experiments to validate (or refute) hypotheses raised in the first version of the manuscript. The structure of the manuscript has improved. The new discussion of the study by Suskiewicz et al. is an important addition. (On that note, however, the authors state that they carried out "...extensive mutagenesis combined with functional studies, which were absent in the previous study, but are fulfilled in our study." I would like to point out that Suskiewicz et al. did in fact carry out mutagenesis followed by ADP-ribosylation assays.) The newly added quantitative characterisation of mutant HPF1 binding by ITC nicely complements the ADP-ribosylation assays in the previous work and this study.

The R239A mutation has been introduced as suggested, and based on these new experiments, the authors' model for activation of the acceptor serine has been revised extensively. Notably, Arg239 is no longer expected to play a direct role in catalysis but rather in providing structural stability (supported by ADP-ribosylation mapping data), in indirectly promoting PARP1 binding, and in positioning Glu284 for catalysis. There is a large amount of new discussion about the roles of Arg239 at the end of the results section. Interestingly, the new data suggest that role of Arg239 may be different than the role proposed by Suskiewicz et al., which will certainly be of interest to the field. In the final revisions, I would advise the authors to avoid the impression of pitching their study against the previous one, even if this was not intended; in fact, the two studies nicely complement each other in many ways.

As pointed out below, the clarity of the message of the manuscript could be further improved.

Almost all of the minor concerns about formatting and phrasing have been addressed. Labelling of figures has improved, and figure legends have improved detail overall, although some more information could still be provided for some (particularly for Supplementary Figures 2, 3 and 7).

Specific points:

- p. 4, paragraph 2: The crystallographic data alone cannot resolve the question of stoichiometry as both crystallographic and non-crystallographic symmetry could be taken into account to make different arguments. Unequivocal support of the true PARP1:HPF1 interface comes from the mutational analysis informed by the X-ray crystallography data.
- Fig. 2a: E284 is mutated in Fig. 2b but not shown in panel a.
- Fig 2b: The grey arrow should be pointing to the HPF1 E284A band, but instead points to the C285H band.
- p. 5, paragraph 2: I suggest stating: "... did not interfere with PARP function and binding", rather than just "binding", to reflect the two different assays employed.
- p. 5/6: I suggest stating "... joint active site that would potentially accommodate the access of the positively charged histones." This still is and remains a hypothesis.
- p. 6, paragraph 1: There is a lot of discussion of a "tunnel"; however, the "tunnel" is never shown.
- p. 6, paragraph 2: Compared to the previous version of the manuscript, this section has lost clarity. Without consulting the previous version, I would have had great difficulties following. The sentences are very long, and it is unclear which structures and distances are compared with each other. This definitely needs to be addressed. The structural comparison needs to be explained much more clearly.
- p. 9, heading "Why is HPF Arg239 important for HPF1/PARP1 binding?" – This question will almost certainly confuse the reader, as in the previous section the authors have just established that the R239A mutation "only mildly weakens the binding affinity between HPF1 and PARP1".
- p. 9, paragraph 1: "acidic residues (say Asp239 and Glu240)" – did the authors mean to refer to Asp235?
- The PARP1 Glu756 mutant is referred to as E756A in the text, but as D756A in Supplementary Figure 2. ITC data on this mutant is also only introduced in the discussion, perhaps it should also be introduced earlier in the results section.
- Whilst E988 is now shown in Figure 3, there is still no discussion around its role in catalysis.
- I can see why Fig. 3c and d are shown with the same view, but the authors could consider showing different views to improve the clarity of the individual structural representations.
- There are a large number of stylistic issues in the paper (*examples below). Whilst it is beyond my role as a reviewer to provide editing services, I suggest the Editorial Team work closely with the authors to address these. This study makes an important contribution to the field, and it would be unfortunate to allow stylistic and clarity issues distract from that. With regards to style, the manuscript could also benefit from more focus in the discussion.

* A few examples for stylistic issues to be addressed:

- Abstract – "structural insights obtained in the HPF1/PARP2 study"  "structural insights obtained in a recent HPF1/PARP2 study"
- "how HPF1 binding to PARP2 and promoting serine ADPr"  "how HPF1 binding to PARP2 promotes serine ADPr"
- "25.41% of the sequences are identical"  "25.41% of the residues are identical" (Also, a single integer would suffice.)
- "therefore it is unknown whether the tunnel still maintains"  "therefore it is unknown whether the tunnel is maintained"
- "pretty high sequence homology"
- "band mysteriously became weaker"
- "still bound PARP1 pretty well"
- active voice / passive voice, different tenses used, etc.

Overall, the revised version highlights aspects that are unique relative to the Suskiewicz et al. study. With further revision, I support publication of this work.

Reviewer #3 (Remarks to the Author):

The authors have addressed the critical comments of the reviewers in a satisfactory manner. The manuscript has been greatly improved with the new experimental data. However, it would benefit from proof-reading to remove spelling and phrasing errors.

The authors need to clarify in the abstract what they refer to when mentioning the "HPF1/PARP2/ study", namely the paper by Suskiewicz et al.

In supplemental figure S2 the ITC data showing the interaction between HFP1 WT to PARP1 WT with DNA the stoichiometry measured is $N=0.51$ and $N=0.03$, respectively. Can the authors describe why this is the case when we know the stoichiometry is 1:1?

Point-by-point responses to the reviewers' comments

Reviewer #1 (Remarks to the Author):

Sun et al present the crystal structures of HPF1 and the HPF1:PARP1 complex, and a brief biochemical characterization that strengthens the mode of interaction and PARP1 activation/re-direction of specificity.

It is problematic that the authors remain closely attached - in terms of study content, scientific hypotheses, and methodology - to a previous publication (Suskiewicz et al., Nature 579: 598-602) with very similar content and results. One exception are the new ITC data; unfortunately, the authors use ITC only to confirm what we already knew about the complex. Throughout the text the authors stress the small differences to the previous paper: The present study shows the PARP1 (not PARP2) complex with HPF1, and with better resolution than before. Nevertheless, given the high degree of homology between the two enzymes, especially within the active site; and given the extensive experimental verification carried out in the previous paper, it is clear that the present manuscript only adds a minor increment to our understanding of PARP1 activity during the DNA damage response. In fact, this appears to be the authors' own conclusion (see line 22-24).

Response: We thank the reviewer for carefully reading our manuscript, and we agree that our study is similar to the study by Suskiewicz et al. The major conclusions of these two studies are nearly the same, except that we explored in further detail the role of Arg239, which we think is a useful addition to the Suskiewicz et al study. Although the high similarity of these two studies is disappointing both to us and to the readers, it proves that the Suskiewicz et al. study and ours cross-validate each other, since we did our own studies back to back (please see the next paragraph for details). We now understand that the writing of our paper may have placed too much emphasis on the small differences between the two studies. This is an important comment for us to improve the writing. We have revised this manuscript to address this comment. We hope that the manuscript now reads more comfortably.

We would like to provide more information on the timeline of our study. We solved the HPF1/PARP1-CAT Δ HD structures on 06-22-2019, while Suskiewicz et al. collected the diffraction data set for HPF1/PARP2-CAT Δ HD on 07-19-2019 (according to www.rcsb.org).

Therefore, we actually solved a HPF1/PARP complex structure earlier than Suskiewicz *et al.* and prior to the publication of their paper. Although our structures and findings are highly similar to those of Suskiewicz *et al.*, we carried out our study independently, without any input from the Suskiewicz *et al.* paper. We regret that we finalized our work and manuscript preparation several months later than Suskiewicz *et al.* The reason is that Fa-Hui Sun, the first author of our study, is a graduate student. According to our university policy, we must allow students to write the first version of the manuscript themselves as mandatory training. This slowed down the writing process for several months.

It is unfortunate that the authors chose not to pursue any questions and concepts that were left unanswered by the previous study, including questions they mention in their text. For instance, how can the new findings inform drug discovery (mentioned in several instances, but not addressed by experiment)? Are there any PARP inhibitors that preferentially inhibit either the HPF1 complex or PARP1 alone (i.e., serine ADP-ribosylation vs carboxylic acid ADP-ribosylation)? Similarly, the authors might have chosen to examine the consequences of their and the Ahel group's findings in cells during DNA stress conditions.

In summary, I believe that although this study is timely, the results presented have limited novelty and originality.

Response: Studies on PARP1 inhibitors in the context of HPF1/PARP1 have been planned and are ongoing. We cannot include these data in the current manuscript as it is beyond the scope of the current study (the biology of HPF1/PARP1). We noted that in the Suskiewicz *et al.* paper, the authors also mentioned that "... HPF1 might directly affect PARP inhibition and trapping" without further addressing this point with experiments, likely due to the same considerations. We thank the reviewer for the suggestion of cell line studies under DNA stress conditions. Unfortunately, as structural biologists, we are not able to carry out high-standard cell line studies at this time. We are afraid that as non-professionals in cell research, we might introduce more confusion into the field rather than resolving questions at this time, but we are trying to establish this research capacity, and are open to collaborations in the future.

Reviewer #2 (Remarks to the Author):

I thank the authors for their thorough response to my previous comments! I welcome the extensive analysis of mutant variants and the experiments to validate (or refute) hypotheses raised in the first version of the manuscript. The structure of the manuscript has improved. The new discussion of the study by Suskiewicz et al. is an important addition. (On that note, however, the authors state that they carried out "...extensive mutagenesis combined with functional studies, which were absent in the previous study, but are fulfilled in our study." I would like to point out that Suskiewicz et al. did in fact carry out mutagenesis followed by ADP-ribosylation assays.) The newly added quantitative characterization of mutant HPF1 binding by ITC nicely complements the ADP-ribosylation assays in the previous work and this study.

Response: We thank the reviewer for the positive comments and the kind reminder. We have deleted the "which...but..." clause.

The R239A mutation has been introduced as suggested, and based on these new experiments, the authors' model for activation of the acceptor serine has been revised extensively. Notably, Arg239 is no longer expected to play a direct role in catalysis but rather in providing structural stability (supported by ADP-ribosylation mapping data), in indirectly promoting PARP1 binding, and in positioning Glu284 for catalysis. There is a large amount of new discussion about the roles of Arg239 at the end of the results section. Interestingly, the new data suggest that role of Arg239 may be different than the role proposed by Suskiewicz et al., which will certainly be of interest to the field. In the final revisions, I would advise the authors to avoid the impression of pitching their study against the previous one, even if this was not intended; in fact, the two studies nicely complement each other in many ways.

Response: We thank the reviewer for the kind comment and suggestions. The manuscript has been revised to avoid the impression of pitching our study against the previous one.

As pointed out below, the clarity of the message of the manuscript could be further improved. Almost all of the minor concerns about formatting and phrasing have been addressed. Labelling of figures has improved, and figure legends have improved detail overall, although some more information could still be provided for some (particularly for Supplementary Figures 2, 3 and 7).

Response: We thank the reviewer for the positive comments and suggestions. The whole manuscript, including the legends for Supplementary Figures 2, 3 and 7 has been further revised according to the reviewer's suggestions.

Specific points:

- p. 4, paragraph 2: *The crystallographic data alone cannot resolve the question of stoichiometry as both crystallographic and non-crystallographic symmetry could be taken into account to make different arguments. Unequivocal support of the true PARP1:HPF1 interface comes from the mutational analysis informed by the X-ray crystallography data.*

Response: We thank the reviewer for the observation. We have removed the last sentence in p. 4, paragraph 2 according to the suggestion.

- Fig. 2a: *E284 is mutated in Fig. 2b but not shown in panel a.*

Response: The stick representation of E284 has been added in the Figure 2a.

- Fig 2b: *The grey arrow should be pointing to the HPF1 E284A band, but instead points to the C285H band.*

Response: We thank the reviewer for carefully reading our manuscript and finding this error. The grey arrow has been repositioned to precisely point to the E284A band.

- p. 5, paragraph 2: *I suggest stating: "... did not interfere with PARP function and binding", rather than just "binding", to reflect the two different assays employed.*

Response: We thank the reviewer for their thoughtful suggestion, and the manuscript has been revised accordingly.

- p. 5/6: I suggest stating "... joint active site that would potentially accommodate the access of the positively charged histones." This still is and remains a hypothesis.

Response: We agree with the reviewer. The word "potentially" has been added to make the statement more accurate.

- p. 6, paragraph 1: *There is a lot of discussion of a “tunnel”; however, the “tunnel” is never shown.*

Response: We thank the reviewer for pointing this out. We have re-drawn Figure 3b. In the new version, the HD subdomain is shown as transparent (to visualize the cartoon presentation of HD and the sticks of the key residues) and yellow (to distinguish from the surface potential presentation of HPF1) surface. The entrance of the tunnel can now be seen clearly, and the benzamide molecule in the active center can be seen through the entrance of the tunnel. We also placed a green arrow to indicate the entrance, for clarity.

- p. 6, paragraph 2: *Compared to the previous version of the manuscript, this section has lost clarity. Without consulting the previous version, I would have had great difficulties following. The sentences are very long, and it is unclear which structures and distances are compared with each other. This definitely needs to be addressed. The structural comparison needs to be explained much more clearly.*

Response: We thank the reviewer for the helpful comments. The paragraph has been revised and the corresponding figure panels have also been remade to improve the clarity.

- p. 9, heading “*Why is HPF Arg239 important for HPF1/PARP1 binding?*” – *This question will almost certainly confuse the reader, as in the previous section the authors have just established that the R239A mutation “only mildly weakens the binding affinity between HPF1 and PARP1”.*

Response: We agree with the reviewer's opinion and the heading has been changed to “Indirect contribution of Arg239 to HPF1/PARP1 binding”.

- p. 9, paragraph 1: *“acidic residues (say Asp239 and Glu240)” – did the authors mean to refer to Asp235?*

Response: We thank the reviewer for very careful reading and kindly pointing out this mistake. We have now corrected the text accordingly.

- *The PARP1 Glu756 mutant is referred to as E756A in the text, but as D756A in Supplementary Figure 2. ITC data on this mutant is also only introduced in the discussion, perhaps it should also be introduced earlier in the results section.*

Response: We thank the reviewer for pointing out this mistake. The 756th amino acid residue of

PARP1 is indeed Asp, and we have corrected this error in the text.

- *Whilst E988 is now shown in Figure 3, there is still no discussion around its role in catalysis.*

Response: We thank the reviewer for carefully checking the figures. Since we mention Glu988 in the main text of the revised manuscript (please see section “*HPF1 binding remodels the PARP1 active site for histone serine ADP-ribosylation*”), we have decided to keep E988 in Figure 3.

- I can see why Fig. 3c and d are shown with the same view, but the authors could consider showing different views to improve the clarity of the individual structural representations.

Response: We thank the reviewer for the constructive suggestion. We have replaced Figure 3.d with a different view to improve the clarity.

- *There are a large number of stylistic issues in the paper (*examples below). Whilst it is beyond my role as a reviewer to provide editing services, I suggest the Editorial Team work closely with the authors to address these. This study makes an important contribution to the field, and it would be unfortunate to allow stylistic and clarity issues distract from that. With regards to style, the manuscript could also be benefit from more focus in the discussion.*

** A few examples for stylistic issues to be addressed:*

- *Abstract – “structural insights obtained in the HPF1/PARP2 study”  “structural insights obtained in a recent HPF1/PARP2 study”*

- *“how HPF1 binding to PARP2 and promoting serine ADPr”  “how HPF1 binding to PARP2 promotes serine ADPr”*

- *“25.41% of the sequences are identical”  “25.41% of the residues are identical” (Also, a single integer would suffice.)*

- *“therefore it is unknown whether the tunnel still maintains”  “therefore it is unknown whether the tunnel is maintained”*

- *“pretty high sequence homology”*

- *“band mysteriously became weaker”*

- *“still bound PARP1 pretty well”*

- *active voice / passive voice, different tenses used, etc.*

Response: We thank the reviewer for these helpful suggestions. The above issues have been

resolved as suggested, and the whole paper has been proofread by a professional scientific English editing company.

Overall, the revised version highlights aspects that are unique relative to the Suskiewicz et al. study. With further revision, I support publication of this work.

Reviewer #3 (Remarks to the Author):

The authors have addressed the critical comments of the reviewers in a satisfactory manner. The manuscript has been greatly improved with the new experimental data. However, it would benefit from proof-reading to remove spelling and phrasing errors.

The authors need to clarify in the abstract what they refer to when mentioning the "HPF1/PARP2/ study", namely the paper by Suskiewicz et al.

Response: We have now rephrased the sentence as "... obtained in a recent HPF1/PARP2 study by Suskiewicz et al. apply to..." to clarify the reference to the "HPF1/PARP2/ study".

In supplemental figure S2 the ITC data showing the interaction between HPF1 WT to PARP1 WT with DNA the stoichiometry measured is $N=0.51$ and $N=0.03$, respectively. Can the authors describe why this is the case when we know the stoichiometry is 1:1?

Response: We thank the reviewer very much for carefully reading our manuscript. It has become clear that HPF1 binding to full-length PARP1 depends on DNA binding. Since DNA reversibly binds to PARP1, it is reasonable to infer that only a certain amount of PARP1 molecules were bound to DNA and became activated in the titration, and only this portion of activated PARP1 molecules were able to bind HPF1. This explains why the N number is only about 0.5 for both PARP1 WT and the D756A mutant. The $N=0.03$ titration of PARP1 WT was done in a final experiment. The extremely low N number indicates that the true concentration of HPF1-bindable PARP1-DNA complexes was much lower than the expected value. After checking our experimental records, we found that this titration was done several hours after the first one (the $N=0.51$ one). Therefore, it is likely that when we did the $N=0.03$ titration the

protein sample had undergone partial degradation after storage in solution for a long time. In the revised manuscript we have removed the N=0.03 titration to avoid the confusion.